# Designing Comprehensive Shifting Control Strategy of Hydro-Mechanical Continuously Variable Transmission

**Jiang Li** [1,2], **Hao Dong** [3], **Bing Han** [1,2], **Yanan Zhang** [1,2] **and Zhongxiang Zhu** [1,2,*]

1    Beijing Key Laboratory of Optimized Design for Modern Agricultural Equipment, College of Engineering, China Agricultural University, Beijing 100083, China; jiangli_cau@163.com (J.L.); hanbing981114@163.com (B.H.); 15063511839@163.com (Y.Z.)
2    The Soil-Machine-Plant Key Laboratory of the Ministry of Agriculture of China, Beijing 100083, China
3    State Key Laboratory of Power System of Tractor, Luoyang 471039, China; dhao@ytjszx.com
*    Correspondence: zhuzhonxiang@cau.edu.cn

**Abstract:** The shift quality is one of the most important criteria for evaluating the operation and transportation of a hydro-mechanical continuously variable transmission (HMCVT) tractor. In this study, a comprehensive shifting control strategy of a hydro-mechanical continuously variable transmission in a high horsepower tractor is investigated for enhancing shift quality. The detailed mathematical model of the transmission assembly components, such as the pump-motor system, the clutch system and the equivalent gear shaft system of the hydro-mechanical continuously variable transmission, are established to analyze the exact shifting process. For different transmission systems, two different control methods are proposed to achieve precise control of the transmission. The double-loop control (PID-Model Predictive Control) is used for the pump-motor hydraulic system, which not only ensures the anti-interference of the control signal when the displacement is a fixed value but also improves the response characteristics and stability of the control output when the displacement ratio is variable. The PID control is used for oil pressure control of the clutch system, and experimental results reveal that a simple method can obtain better control performance. Finally, a comprehensive control strategy in the process of shifting is proposed, and the optimal control strategy is obtained by comprehensively adjusting the pump-motor hydraulic system and the clutch system at the shifting point. The results show that the shift acceleration under the optimal control strategy is 64.4% lower than that without control. The proposed control strategy can effectively reduce shift shock and Improve shift smoothness.

**Keywords:** shift quality; hydro-mechanical continuously variable transmission; double-loop control (PID-Model Predictive Control); pump-motor hydraulic system; comprehensive shifting control strategy; tractor

## 1. Introduction

The hydro-mechanical continuously variable transmission (HMCVT) has both hydrostatic and mechanical transmission advantages. It can not only realize the continuous change of the speed ratio but also show excellent transmission characteristics and high-efficiency characteristics [1,2]. The HMCVT is widely used in the transmission systems of off-road vehicles (ORV), such as armored vehicles, engineering vehicles and high-power tractors [3]. The transmission form of HMCVT is mainly divided into single planetary transmission and multi-planetary transmission [4], but the problem of shift control has always been a research hotspot, whatever the transmission structure is. At present, contributions to this field can be primarily categorized as follows:

### 1.1. Research on Influencing Factors of Shift Quality

The Caterpillar company studied the shift quality of hydrostatic power split CVT [5], which calculated the optimal shift time by predicting the time when the transmission

speed reaches the shift point. Wang studied the effect of the time and voltage parameters of the clutch on the shift quality of the starting stage of the vehicle [6], and a matching rule for the control parameters was proposed, which revealed the influence weights of different parameters on the shift quality. Similarly, shift control strategies based on physical parameters and shift time are also proposed for four typical operating conditions. The orthogonal testing method (including range and variance analysis method) is proposed to optimize shift control factors which define deceleration amplitude, dynamic load coefficient, maximum shock, and shift time as evaluation indicators [7].

### 1.2. Design and Research of Clutch Shift Controller

Since the widespread application of control theory in the vehicle industry, clutch shift controllers have also been designed with different control methods. Sergio developed a power-split continuously variable transmission (CVT) control system for high-power tractors [8]. The hydraulic transmission ratio servo controller and synchronizer are experimentally verified and evaluated, and the speed ratio tracking performance during section change is improved by increasing the bandwidth.

The integral linear quadratic regulator is introduced into the dynamic model of the clutch. The relative speed difference between the engine and the slipping clutch is adjusted to realize the speed control of the clutch in the inertia phase [9]. Two robustness-based shift control strategies are proposed to optimize the action mechanisms of clutch engagement [10,11]. A controller operates the torque of the torque phase to improve the shift robustness, which controls the electro-hydraulic proportional valve with a different duty cycle in the filling phase and adaptively. Another control method uses three controllers (a feedforward controller, a feedback controller and a disturbance controller) to control the three stages of the shift separately, which improves driving comfort.

Summarizing the technical developments and current trends for solving shift control of HMCVT exposes the following problems.

First, only the condition of the clutch is considered in the shifting process, and the influence of the pump-motor hydraulic system on the shifting process is not considered for the speed change. From the transmission principle of HMCVT (Figure 1), it can be known that the output speed of the shift is composed of two parts: mechanical transmission and hydraulic transmission. Therefore, it presents inaccurate information about the shifting quality as it only considers the constant speed of the mechanical transmission and ignores the influence of the speed of the hydraulic transmission. Secondly, many scholars have studied shift control, but the combined control of the pump-motor hydraulic system and clutch is rarely studied in shifting. More importantly, the pump-motor hydraulic system is a nonlinear load system [12]. Due to the inherent dynamics of hydrostatic transmissions and their input limitations, hydraulic system control suffers from delay issues [8]. Thus, the hydraulic transmission speed is required to change accurately and stably during the shifting process, which puts forward requirements for the anti-interference ability of the pump-motor system.

Therefore, this study proposes different control methods for the pump-motor hydraulic system and the clutch system of the HMCVT. Firstly, according to the characteristics of the pump-motor hydraulic system, a new control method (PID-MPC double-loop control) has been designed, which improves the hydraulic system's speed tracking and anti-interference characteristics. The PID control is used to control the oil pressure of the clutch, which not only has a simple control process but also excellent control precision. In order to optimize the shifting process, a detailed mathematical model of the key system is established, and a shifting control strategy with the best shifting quality is explored under the comprehensive control conditions. The results show that the control strategy effectively improves the shifting quality of the high-horsepower tractor with HMCVT.

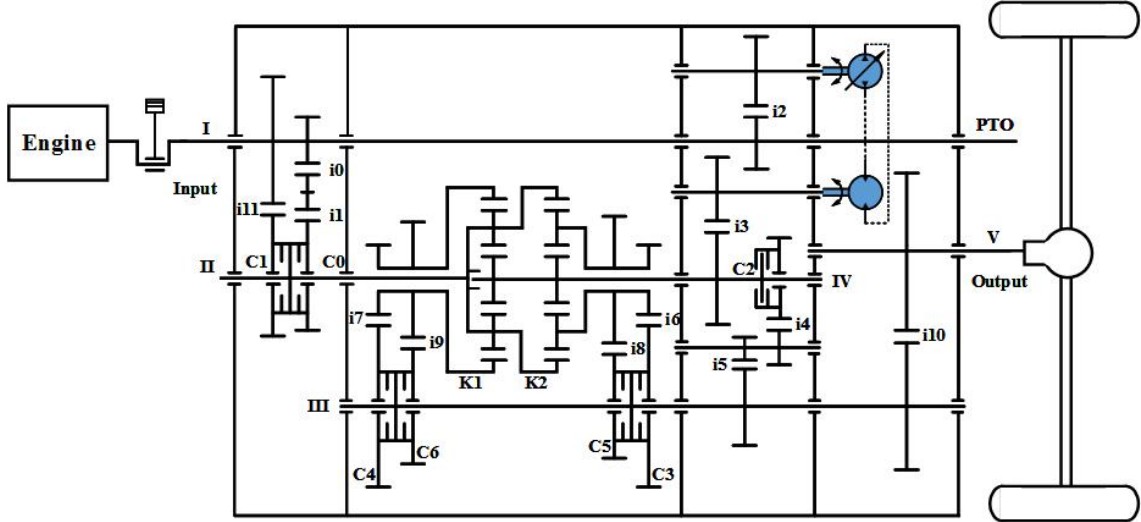

**Figure 1.** Transmission principles of an HMCVT (I, II, III, I, II, III, IV, V represent the different drive shafts; C1,C2, C3, C4 represent different clutches; i1, i2, . . . , i11 represent different gear pairs).

## 2. Mathematical Modeling and Methods

The materials and methods are described with sufficient detail to allow others to replicate and build on the published results. Please note that the publication of your manuscript implicates that you must make all materials, data, computer code, and protocols associated with the publication available to readers. Please disclose at the submission stage any restrictions on the availability of materials or information. New methods and protocols should be described in detail, while well-established methods can be briefly described and appropriately cited.

The dynamic model of HMCVT mainly includes the pump-motor hydraulic system model, valve-clutch model and mechanical shaft model (Figure 1) [13]. The engine power is divided into two parts through the input shaft I; one part is transmitted to the pump-motor hydraulic system through the gear pair i2; the other part is transmitted to the intermediate shaft II through the clutch C0. Then the two parts of power are combined in the double planetary row structure K1–K2 and transmitted to the output shaft VI.

### 2.1. Pump-Motor Hydraulic System Model

The time-domain mathematical model of the hydraulic pump-motor system is established in this study. This hydraulic system consists of an electro-mechanical conversion unit (EM), a variable piston-variable pump swash plate inclination control unit (PPSP), and a variable pump controlling a fixed motor loop (PM). The dynamic equations of each part are established (Equations (1)–(7)), and then the time-domain mathematical model of the system is synthesized. The principle is shown in Figure 2.

Assumptions: the connecting pipelines between the pump and the motor are very short; the pressure loss and the pipeline dynamics are simplified; the leakage of the pump and motor is laminar flow; the shell pressure of the pump and motor is atmospheric pressure; and the leakage of the low-pressure cavity to the shell is ignored [14].

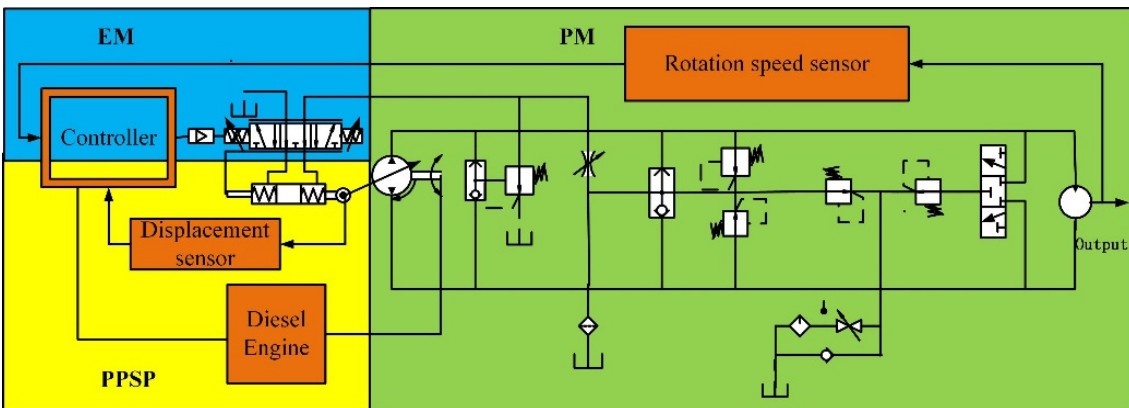

**Figure 2.** Working principle diagram of variable pump control fixed motor.

### 2.1.1. Electro-Mechanical Conversion Unit

The electro-mechanical conversion unit is an important part of the hydraulic control system [15]. Its main function is to convert the input electrical signal into a mechanical scalar (force and displacement). As with the previous stage of electro-hydraulic control components, it needs rich steady-state control accuracy, dynamic response performance, anti-interference ability and working reliability [16]. The electro-mechanical conversion unit model is expressed as follows:

$$
\begin{cases}
F_M = K_I I - K_Y x_v \\
F_M = m_t \frac{d^2 x_v}{dt^2} + \beta_t \frac{dx_v}{dt} + T \\
K_y = \frac{\partial F_M}{\partial y} + K_{sy} \\
F_x = T = m_v \frac{d^2 x_v}{dt^2} + \beta_v \frac{dx_v}{dt} + K_V x_v + K_{fV} x_v
\end{cases}
\tag{1}
$$

where $F_M$ represents the thrust force of the electromagnet; $x_v$ represents the armature stroke; $K_I$ represents the current force gain of the proportional electromagnet; $K_Y$ represents the sum of the displacement force gain of the proportional electromagnet and the stiffness of the zero-adjusting spring; $m_t$ represents the mass of the armature assembly; $\beta_t$ represents the damping coefficient; $m_v$ represents the quality of the spool; $x_v$ represents displacement of the spool.; $F_x$ represents the force between armature and valve core; $\beta_v$ represents the viscous damping coefficient of the spool; $K_V$ represents the spring stiffness of the spool; $K_{fV}$ represents steady-state hydrodynamic stiffness coefficient.

### 2.1.2. Control Unit of the Swash Plate Inclination

The control unit of the inclination angle of the swash plate is composed of two parts: a valve-piston cylinder unit and a piston-swash plate inclination unit.

The dynamic balance equation of the valve-piston cylinder unit is shown as follows:

$$
\begin{cases}
Q_L = K_q x_c - K_c P_L \\
Q_L = A_p \frac{dy}{dt} + C_{tp} P_L + \frac{V_t}{4\beta_e} \frac{dP_L}{dt} \\
A_P P_L = M_t \frac{d^2 y}{dx} + B_p \frac{dy}{dx} + Ky + F_L
\end{cases}
\tag{2}
$$

where $Q_L$ represents the load flow; $K_q$ represents flow amplification factor; $K_C$ represents the flow pressure coefficient; $x_c$ represents the spool displacement; $P_L$ represents the load pressure difference; $A_p$ represents the effective area of hydraulic cylinder piston; $V_t$ represents the volume of the hydraulic cylinder; $y$ represents the piston displacement; $C_{tp}$ represents the total leakage coefficient of the hydraulic cylinder; $M_t$ represents the total mass of the piston and the load; $B_p$ represents the viscous damping coefficient of the piston and the load; $K$ represents the spring stiffness of the piston and the load; $F_L$ represents the load force on the piston.

### 2.1.3. Piston-Swash Plate Inclination

In the pump-motor system, the pump displacement is adjusted by the swash plate inclination, which is controlled by the point proportional valve. The piston-swash plate inclination angle mechanism is essentially a position control system. The position of the variable piston in the mechanism corresponds to the displacement ratio of the pump-motor [12]. So, the unit can be expressed as:

$$y = R \cdot \sin \gamma \tag{3}$$

### 2.1.4. Variable Pump-Fixed Motor System

The variable pump-fixed motor system is the main executive part of the hydraulic system. The PM could adjust the output speed of the hydraulic system by the displacement ratio. The mathematical model is as follows:

$$\begin{cases} D_p = K_p \gamma \\ q_p = D_p \omega_p - C_{ip}(p_1 - p_r) - C_{ep} p_1 \\ q_p = C_{im}(p_1 - p_r) + C_{em} p_1 + D_m \frac{d\theta_m}{dt} + \frac{V_0}{\beta_e} \frac{dp_1}{dt} \\ D_m(p_1 - p_r) = J_t \frac{d^2\theta_m}{dt^2} + B_m \frac{d\theta_m}{dt} + G\theta_m + T_L \end{cases} \tag{4}$$

where $K_p$ represents the displacement gradient of the variable pump; $D_p$ represents the displacement of the variable pump; $\gamma$ represents the swash plate swing angle of the variable mechanism; $q_p$ represents the output flow of the variable pump; $\omega_p$ Represents the speed of the variable pump; $C_{ip}$ represents the internal leakage coefficient of the variable pump; $C_{ep}$ represents the external leakage coefficient of the variable pump; $p_1$ represents the pressure on the high pressure chamber side; $p_r$ represents the low pressure pipeline pressure; $\theta_m$ represents the shaft rotation angle; $C_{im}$ represents the internal leakage coefficient of the motor; $C_{em}$ represents the external leakage coefficient of the motor; $V_0$ represents the working chamber of the pump and the motor and the total volume of the connecting pipe; $D_m$ represents the motor displacement; $\beta_e$ represents the effective volume the amount of elastic film; $J_t$ represents the total moment of inertia of the hydraulic motor and the load (converted to the motor shaft); $B_m$ represents the total viscous damping coefficient of the hydraulic motor and the load (converted to the motor shaft); $G$ represents the torsional spring stiffness of the load; $T_L$ represents the acting on the motor external load torque on the shaft.

### 2.1.5. Speed Sensor and Proportional Amplifier

In the pump-motor hydraulic system, the speed sensor is mainly used to feedback the motor speed signal. Therefore, the speed sensor can be simplified as a proportional link.

$$U = \omega_m K_V \tag{5}$$

The proportional amplifier is represented by a proportional link, and its mathematical model is as follows:

$$K_a = \frac{I}{U} \tag{6}$$

### 2.2. Mathematical Model of Clutch System

The clutch system model is mainly composed of an electro-hydraulic proportional valve and a clutch.

### 2.2.1. Electro-Hydraulic Proportional Valve Model

The leakage between the main spool of the proportional valve and the valve sleeve is too small to be ignored. The hydraulic oil flows into the main valve cavity, which compensates for the amount of oil compression in the main valve cavity, and the rest flows into the clutch cylinder.

When the solenoid valve is not energized, the main spool is located at the left end of the proportional valve. The oil in the clutch cylinder flows back to the oil tank through the oil discharge port of the proportional valve, and the clutch is in a disengaged state. When the clutch is energized, the electromagnetic force generated by the solenoid valve drives the spool to move to the right. Then the oil discharge port is gradually closed and the oil inlet port is opened. Finally, the oil flows into the wet clutch cylinder through the throttle port and the clutch gradually engages. The electro-hydraulic proportional valve model mainly consists of the spool force balance equation, the pressure-flow balance equation and the flow continuity equation.

$$
\begin{cases}
m_v \ddot{x}_v + C_v \dot{x}_v + k_p x_v + 2C_d C_v A_e \cos \theta_s p_o \\
= K_i i - K_b x_v + p_o A_Z \\
q_s = \begin{cases}
\text{sign}(P_s - P_c) \, C_V A_i \sqrt{\frac{2(P_s - P_c)}{\rho}}, x_v > S \\
0, x_v = S \\
-C_V A_e \sqrt{\frac{2(P_c - P_e)}{\rho}}, x_v < S
\end{cases} \\
q_s - q_l = \frac{V_{S0}}{\beta_e} \dot{p}_c
\end{cases}
\tag{7}
$$

where $m_v$ is the quality of the spool; $x_v$ is the displacement of the spool; $C_v$ is the viscous damping coefficient of the proportional valve spool; $k_p$ is the return spring stiffness; $C_d$ is the flow coefficient of the orifice; $C_v$ is the flow rate coefficient of the main spool of the proportional valve; $A_Z$ represents the proportional valve spool action area; $A_e$ represents the throttling area of the proportional valve oil discharge port; $A_i$ represents the inlet throttle area; $\theta_s$ represents the jet angle of the main valve throttling port of the proportional valve; $p_o$ represents the proportional valve outlet pressure; $K_i$ represents the driving coefficient of the electromagnet; $K_b$ represents the speed back EMF coefficient; $P_s$ represents the oil supply pressure, $P_c$ represents the outlet pressure of the proportional valve; $S$ represents displacement of the spool from the initial position to the full closure of the drain hole; $\rho$ represents the oil fluid density; $q_s$ represents the flow of the proportional valve; $q_l$ represents the flow of the clutch cylinder, $V_{s0}$ represents the volume of the main valve cavity of the proportional valve, and $\beta_e$ represents the volume elastic modulus of the oil quantity.

### 2.2.2. Wet Clutch Model

During the engagement process of clutch, the simplified content is defined as follows: the oil passage resistance and the influence of oil passage internal leakage are ignored; the leakage of the gap which is between the clutch piston and the seal ring is ignored. The hydraulic oil which flows into the clutch cylinder by the proportion-al valve is divided into three parts: one part fills the cylinder; one part compensates for the oil compression volume; the other part flows out of the clutch cylinder discharge port. The dynamic model of the clutch system is shown as follows:

$$
\begin{cases}
m_p \ddot{x}_p + c_p \dot{x}_p + k_p x_p = F_s + F_d \\
q_l = \text{sign}(P_0 - P_l) C_d \frac{\pi d_l^2}{4} \sqrt{\frac{2|P_0 - P_l|}{\rho}} \\
q_l = A_l \dot{x}_l + \frac{V_{l0} + A_l x_l}{\beta_e} \dot{p}_l \\
F_s = \pi (R_1^2 - R_2^2) P_l \\
F_d = \frac{\pi \rho}{4} (R_1^2 - R_2^2)(R_1^2 + R_2^2 - 2R_{oil}^2)(\alpha \omega_e)^2
\end{cases}
\tag{8}
$$

where the $x_p$ represents displacement of piston; $m_p$ represents clutch equivalent mass; $c_p$ represents piston viscous damping coefficient; $k_p$ represents piston return spring stiffness; $F_s$ represents friction plate pressing force; $F_d$ represents piston centrifugal oil pressure force; $q_l$ represents clutch cylinder flow; $p_0$ represents the clutch inlet pressure; $p_l$ represents the clutch cylinder pressure; $d_l$ represents the diameter of the oil cylinder inlet; $A_l$ represents the piston area; $V_{l0}$ represents the initial volume of the clutch cylinder; $R_1$ represents the inner radius of the clutch piston; $R_2$ represents the outer radius of the clutch piston;

$R_{oil}$ represents the rotation radius of the oil inlet of the clutch; $\alpha$ represents the hysteresis coefficient; $\omega_e$ represents the angular velocity of the clutch cylinder.

### 2.3. Transmission Shaft Model

The HMCVT transmission system consists of a transmission shaft and a planetary row, which can be simplified as shown in Figure 3. According to the transmission principle of HMCVT, the dynamic equation is expressed as Equation (9) [7].

$$\begin{cases} J_1\dot{\omega}_1 + D_1\omega_1 = T_1 - \dfrac{T_{C1}}{i_{11}} - \dfrac{T_{C0}}{i_1 i_0} - \dfrac{T_P}{i_2} \\ J_2\dot{\omega}_2 + D_2\omega_2 = T_S - T_h/(i_4 i_5) \\ J_3\dot{\omega}_3 + D_3\omega_3 = \sum\limits_{n=3}^{6} T_{Cn} + T_h - T_{OUT}/i_{10} \\ J_s\dot{\omega}_s + D_s\omega_s = T_M + T_s - T_h/(i_4 i_5) \\ J_c\dot{\omega}_c + D_c\omega_c = T_{C0} + T_{C1} - \dfrac{T_{C3}}{i_6} - \dfrac{T_{C5}}{i_8} \\ J_r\dot{\omega}_r + D_r\omega_r = T_{C0} + T_{C1} - \dfrac{T_{C4}}{i_8} - \dfrac{T_{C6}}{i_9} \end{cases} \quad (9)$$

where $J$ represents the moment of inertia; 1, 2, 3 represent the equivalent axis; $D$ represents the damping coefficient of the equivalent axis; $T$ represents the torque; $C$ represents the clutch; $i_x$ represents the transmission ratio of the $x$ pair of gear pairs; $s$, $c$ and $r$ represent the sun gear, the planet carrier, ring gear, respectively; $p$ represents the pump, $M$ represents the motor; $h$ represents the pure hydraulic oil circuit.

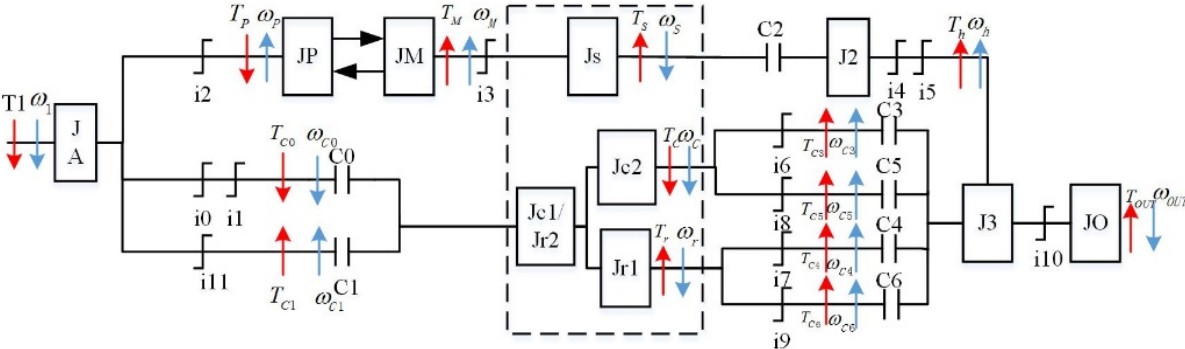

**Figure 3.** Dynamic model of HMCVT shafting (i1, i2, i3, ... i11 represent different gear pairs; P represents the variable pump; M represents the fixed motor).

### 2.4. The Composition of the Control System

The simulation system of HMCVT is shown in Figure 4, which mainly includes hydraulic-mechanical execution system and control system. The mechanical system model is composed of the gear shaft model and the planetary gear train model; the hydraulic system model is composed of a variable pump-fixed motor system and a hydraulic valve-clutch system; the control system of HMCVT is composed of the clutch control system and pump-motor hydraulic control system. The displacement ratio of the variable pump-fixed motor system and the state of the clutch system are controlled by the designed control strategy to realize the accurate management of the transmission ratio and ensure good shift quality.

#### 2.4.1. Design of Clutch Control System

The solenoid proportional valve voltage of the clutch is controlled to guide the operation of the clutch and the feedback pressure is collected and compared with the target pressure. The mathematical model of the clutch system is shown in Section 2.2. The control principle of the clutch system is shown in Figure 5. Some scholars have conducted research on the control method of the clutch. Although the signal tracking error of the model

predictive control and the adaptive control is small, the response speed is slow, which is unfavorable to the control of the clutch for extended periods of time.

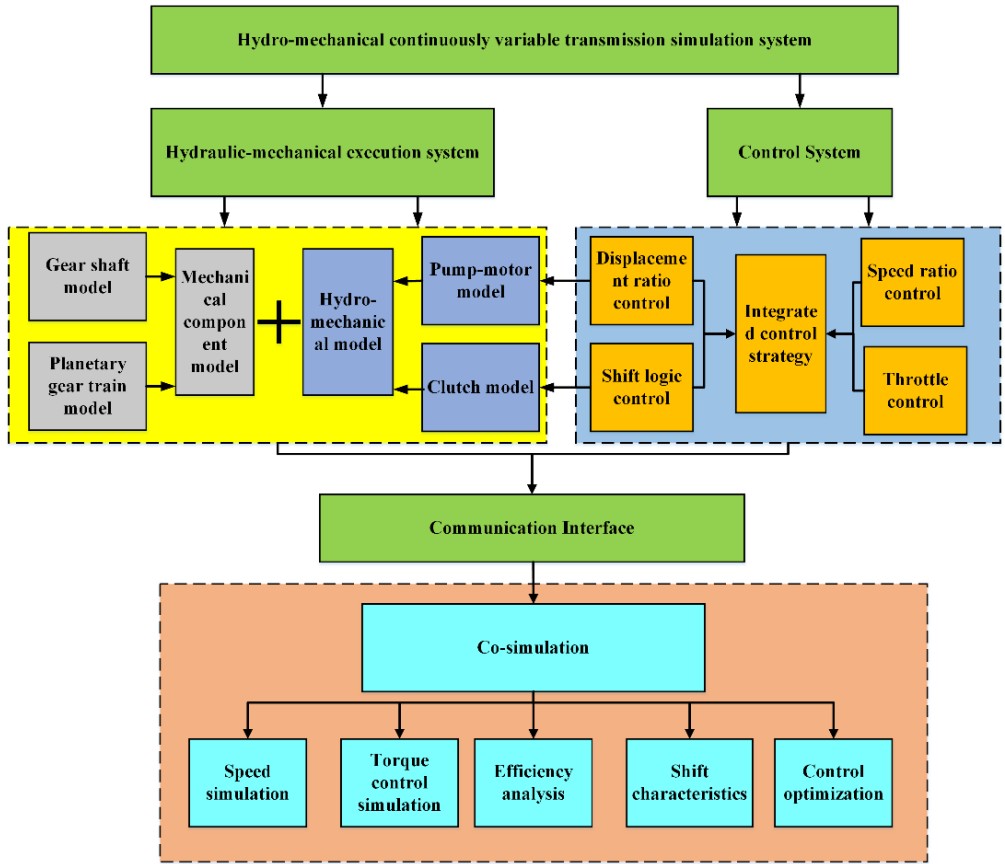

**Figure 4.** HMCVT simulation system diagram.

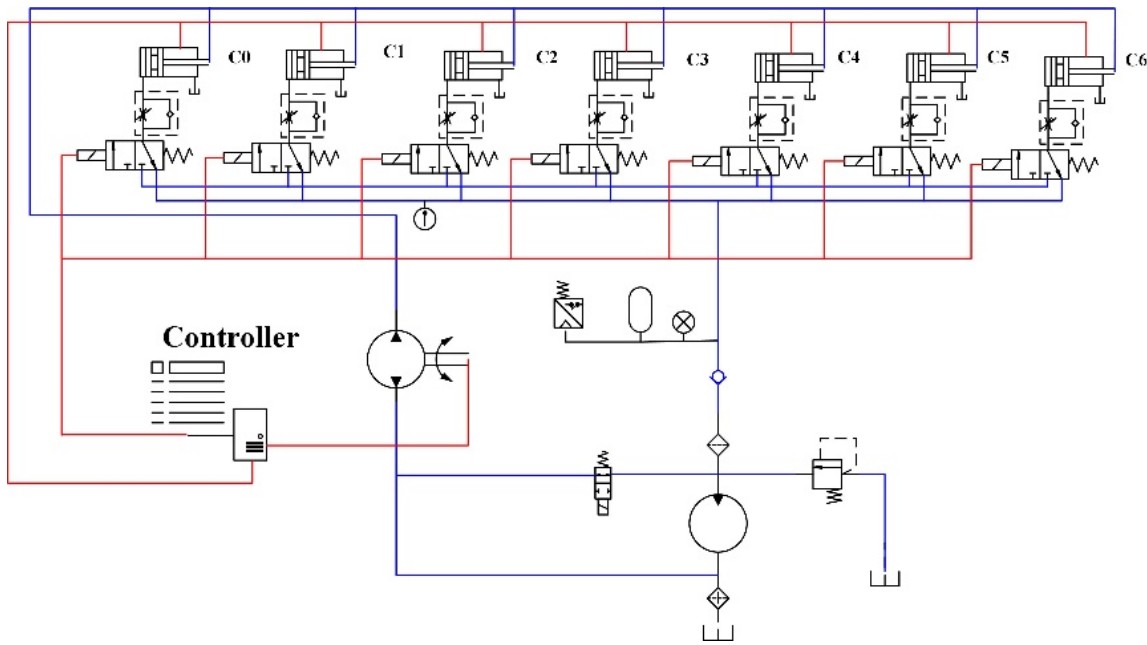

**Figure 5.** Clutch system control principle diagram.

### 2.4.2. Pump-Motor Hydraulic System Control System

The variable pump-fixed motor hydraulic system is the main execution system for speed regulation in the HMCVT stage. The input signal of the proportional valve is controlled by the TCU to adjust the pump-motor displacement ratio. The control principle is shown in Figure 6. The control system is a dual-loop parallel control system. The inner loop adopts PID control, and the outer loop adopts model predictive control (MPC). The inner loop is a servo control of displacement ratio. The displacement of the hydraulic cylinder is fed back to the controller for preliminary adjustment, which is controlled by the electric proportional valve; the function of the outer loop control is the adjustment of the motor output speed. The control input signal is adjusted by comparing the output speed of the feedback motor with the expected value so that the output speed of the motor can reach the ideal value.

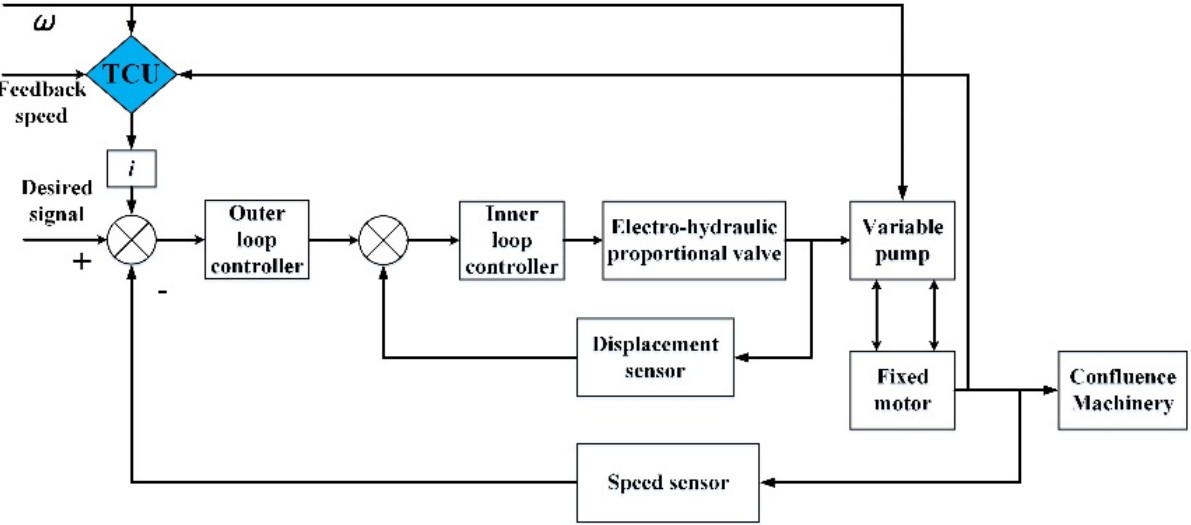

**Figure 6.** Pump-motor hydraulic control system principle diagram.

The constant speed control is an important part of pump-motor speed servo control. Due to the complexity of the flow coupling characteristics of the hydraulic system, the most serious difficulty in achieving constant speed control of the motor is the sudden disturbance of the load torque [17]. The PID control is the most common control algorithm [18]. However, as a result of the very obvious load fluctuations during operation, the hydraulic mechanical equipment will be significantly affected when the engine changes speed. It is very difficult to control the speed ratio parameters of the gearbox to ensure that the engine speed responds quickly and remains stable. Therefore, the role of integral control is not obvious.

The interference from the mechanical transmission parts with the pump-motor hydraulic system makes the HMCVT transmission ratio prediction inaccurate, which results in unstable tractor speed. However, when the pump-motor hydraulic system is in a steady-state, HMCVT's pump-motor hydraulic system needs a faster response speed and better steady-state robustness [19]. Therefore, based on the above characteristics, this study firstly performs model predictive control (MPC) on the outer loop of the input voltage–output speed of the hydraulic system. The displacement equation of the variable pump is:

$$T_P \dot{q}_p + q_p = k_p u_p \tag{10}$$

where $T_P$ represents the torque of the variable pump; up represents the control voltage; $k_p$ represents variable pump displacement control coefficient.

The kinetic Formula (4) is simplified: the low-pressure side is connected to the fuel tank ($p_r = 0$). The equation $n_m = d\theta_m/dt$ can express the motor speed ($n_m$), and the established space state equation is expressed as follows:

$$
\begin{cases}
\dot{q}_p = \frac{1}{T_p}(k_p u_p - q_p) \\
\dot{p} = \frac{\beta_e}{V_0}(q_p - (c_{im} + c_{em})p - D_m n_m) \\
\dot{n}_m = \frac{1}{J_t}(D_m p - B_m n_m + T_L)
\end{cases}
\tag{11}
$$

The state variables are chosen as:

$$
x = \begin{bmatrix} x_1, x_2, x_3, x_4, x_5 \end{bmatrix}^T = [q_p, p, n_m, u_p, T_L]^T
\tag{12}
$$

The control variables as:

$$
U = [u_1, u_2]^T = \begin{bmatrix} \dot{u}_p, \dot{T}_L \end{bmatrix}
\tag{13}
$$

With (4), (11), (12), (13), the dynamic model can be shown as:

$$
\begin{cases}
\dot{x}_1 = -a_1 x_1 + b_1 x_4 \\
\dot{x}_2 = a_2 x_1 - a_3 x_2 - a_4 x_3 \\
\dot{x}_3 = a_5 x_2 - a_6 x_3 + b_2 x_5 \\
\dot{x}_4 = u_1 \\
\dot{x}_5 = u_2
\end{cases}
\Rightarrow
\begin{cases}
a_1 = \frac{1}{T_p}, a_2 = \frac{\beta_e}{V_0}, a_3 = \frac{\beta_e}{V_0}(c_{im} + c_{em}), \\
a_4 = \frac{\beta_e}{V_0} D_m a_5 = \frac{D_m}{J_t}, a_6 = \frac{B_m}{J_t}, \\
b_1 = \frac{k_p}{T_p}, b_2 = \frac{T_L}{J_t}
\end{cases}
\tag{14}
$$

The transmission model of the variable pump-fixed motor is obtained.

$$
\begin{cases}
\dot{x} = Ax + Bu \\
Y = Hx
\end{cases}
\tag{15}
$$

$$
A = \begin{bmatrix}
-\frac{1}{T_p} & 0 & 0 & \frac{k_p}{T_p} & 0 \\
\frac{\beta_e}{V_0} & -\frac{\beta_e}{V_0}(c_{im} + c_{em}) & -\frac{\beta_e}{V_0} D_m & 0 & 0 \\
0 & \frac{D_m}{J_t} & -\frac{B_m}{J_t} & 0 & \frac{T_L}{J_t} \\
0 & 0 & 0 & 0 & 0 \\
0 & 0 & 0 & 0 & 0
\end{bmatrix},
$$

$$
B = \begin{bmatrix}
0 & 0 \\
0 & 0 \\
0 & 0 \\
1 & 0 \\
0 & 1
\end{bmatrix}, \quad
H = \begin{bmatrix} 0 & 0 & 1 & 0 & 0 \end{bmatrix}
\tag{16}
$$

The transmission model is discretized:

$$
\begin{cases}
x(k+1) = A_d x(k) + B_d u(k) \\
y(k) = C_d x(k)
\end{cases}
\tag{17}
$$

Assuming that the prediction time domain of the system is $N_P$ and the control time domain is $N_m$, the predicted value at time $k$ is:

$$
\begin{cases}
x(k+1|k) = A_d x(k) + B_d u(k) \\
x(k+2|k) = A_d x(k+1) + B_d u(k+1) \\
= A_d^2 x(k) + A_d B_d u(k) + B_d u(k+1) \\
\dots \\
x(k+N_p|k) = A_d^{N_p} x(k) + A_d^{N_p-1} B_d u(k) \\
+ A_d^{N_p-2} B_d u(k+1) + \dots + A_d^{N_p-N_m} B_d u(k+N_m-1)
\end{cases}
\tag{18}
$$

The predicted output values of the controlled system are shown:

$$
\begin{cases}
y(k+1|k) = C_d A_d x(k) + C_d B_d u(k) \\
y(k+2|k) = C_d A_d x(k+1) + C_d B_d u(k+1) \\
= C_d A_d^2 x(k) + C_d A_d B_d u(k) + C_d B_d u(k+1) \\
\dots \\
y(k+N_p|k) = C_d A_d^{N_p} x(k) + C_d A_d^{N_p-1} B_d u(k) \\
+ C_d A_d^{N_p-2} B_d u(k+1) + \dots + C_d A_d^{N_p-N_m} B_d u(k+N_m-1)
\end{cases}
\tag{19}
$$

The general formula for the output prediction value of the controlled system in the prediction time domain is:

$$
\begin{cases}
\mathbf{Y}(k) = \mathbf{F}\mathbf{x}(k) + \mathbf{G}\mathbf{U} \\
\mathbf{Y}(k) = \begin{bmatrix} y(k+1|k) & y(k+2)|k & \dots & y(k+N_P)|k \end{bmatrix}^T \\
\mathbf{F} = \begin{bmatrix} C_d A_d & C_d A_d^2 & \dots & C_d A_d^{N_p} \end{bmatrix}^T \\
\mathbf{U} = \begin{bmatrix} u(k) & u(k+1) & \dots & u(k+N_m-1) \end{bmatrix}
\end{cases}
\tag{20}
$$

$$
G = \begin{bmatrix}
C_d B_d & 0 & \dots & 0 \\
C_d B_d A_d & C_d B_d & \dots & 0 \\
\dots & \dots & \dots & \dots \\
C_d B_d A_d^{N_P-1} & C_d B_d A_d^{N_P-2} & \dots & C_d B_d A_d^{N_P-N_m}
\end{bmatrix}
\tag{21}
$$

The difference between the state variable and the output value at time $k+1$ and time $k$ can be obtained.

$$
\begin{cases}
x(k+1) - x(k) = A_d(x(k+1) - x(k)) \\
+ B_d(u(k+1) - u(k)) \\
y(k+1) - y(k) = C_d(x(k+1) - x(k))
\end{cases}
$$
$$
\Downarrow
$$
$$
\begin{cases}
\Delta x(k+1) = A_d \Delta x(k) + B_d \Delta u(k) \\
\Delta y(k+1) = C_d A_d \Delta x(k) + C_d B_d \Delta u(k)
\end{cases}
\tag{22}
$$
$$
\Downarrow
$$
$$
\begin{cases}
x_u(k+1) = A_u x_u(k) + B_u \Delta u(k) \\
y(k) = C_u x_u(k)
\end{cases}
$$

$$
x_u(k) = \begin{bmatrix} \Delta x(k) \\ y(k) \end{bmatrix}; \quad A_u = \begin{bmatrix} A_d & 0_d^T \\ C_d A_d & 1 \end{bmatrix}; \quad B_u = \begin{bmatrix} B_d \\ C_d B_d \end{bmatrix};
$$
$$
F_u = \begin{bmatrix} C_u A_u \\ C_u A_u^2 \\ \dots \\ C_u A_u^{N_p} \end{bmatrix}; \quad C_u = \begin{bmatrix} 0_d^T & 1 \end{bmatrix}
\tag{23}
$$

The performance optimization index function of MPC is:

$$
J = (Y_r - Y)^T R_s(Y - Y_r) + \Delta U^T R \Delta U
\tag{24}
$$

where $Y_r$ represents the expected value matrix; $R$ represents the control weight matrix.

The error weight matrix is:

$$
R_S^T = \underbrace{\begin{bmatrix} 1 & 1 & \dots & 1 \end{bmatrix}}_{N_P}^T y_r(k) = \overline{R}_S^T y_r(k)
\tag{25}
$$

$$
\mathbf{J} = [\mathbf{Y_r} - \mathbf{F_u x_u}(k)]^T \mathbf{R_s}[\mathbf{Y_r} - \mathbf{F_u x_u}(k)] - 2\Delta \mathbf{U^T G^T R}_s \Delta \mathbf{U}[\mathbf{Y_r} - \mathbf{F_u x}(k)]
+ \Delta \mathbf{U^T}(\mathbf{G^T R}_s \mathbf{G} + \mathbf{R})\Delta \mathbf{U}
\tag{26}
$$

$$
\frac{\partial \mathbf{J}}{\partial \Delta \mathbf{U}} = 0 \Rightarrow \Delta \mathbf{U} = (\mathbf{G^T G} + \mathbf{R})^{-1} \mathbf{G^T}(\overline{\mathbf{R}_s} y_r(k) - \mathbf{F_u} x(k))
\tag{27}
$$

The state prediction value of the controlled system at time $K + 1$ is:

$$x_u(k+1) = (\mathbf{A}_u - \mathbf{B}_u K_E)x_u(k) + \mathbf{B}_u K_r y_r(k) = \mathbf{A}_u x_u(k) + \mathbf{B}_u \Delta u(k) \qquad (28)$$

$K_E$ is the first row element of matrix $(\mathbf{GTG} + \mathbf{R}) - \mathbf{1GTFu})$; $K_r$ is the first row element of matrix $(\mathbf{GTG} + \mathbf{R}) - \mathbf{1GT\overline{R}s})$. $K_E$ can be described as $K_E = [K_x\ K_r]$. $K_x$ can be expressed as the feedback gain related to the state variable, and $K_r$ is expressed as the feedback gain which relates to the output of the control system. The system control block diagram of the state extension MPC is shown in Figure 7.

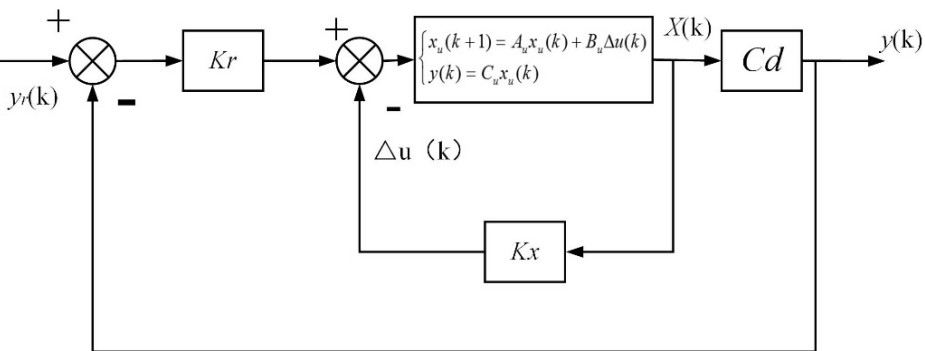

**Figure 7.** System control block diagram of state extension MPC.

### 2.5. HMCVT Control Strategy

The HMCVT shift control strategy mainly consists of a transmission ratio control module, a throttle opening control module, a shift control module and a load feedback module, as shown in Figure 8. The main task of hybrid power system control is to accurately realize the (unknown a priori) motion commands given by the driver through accelerator pedal operation or driving strategy algorithms to control the pump-motor hydraulic system and clutch during gear shifts [20]. The TCU uses an integrated approach to realize a smooth transition of segment change according to the driver's operation.

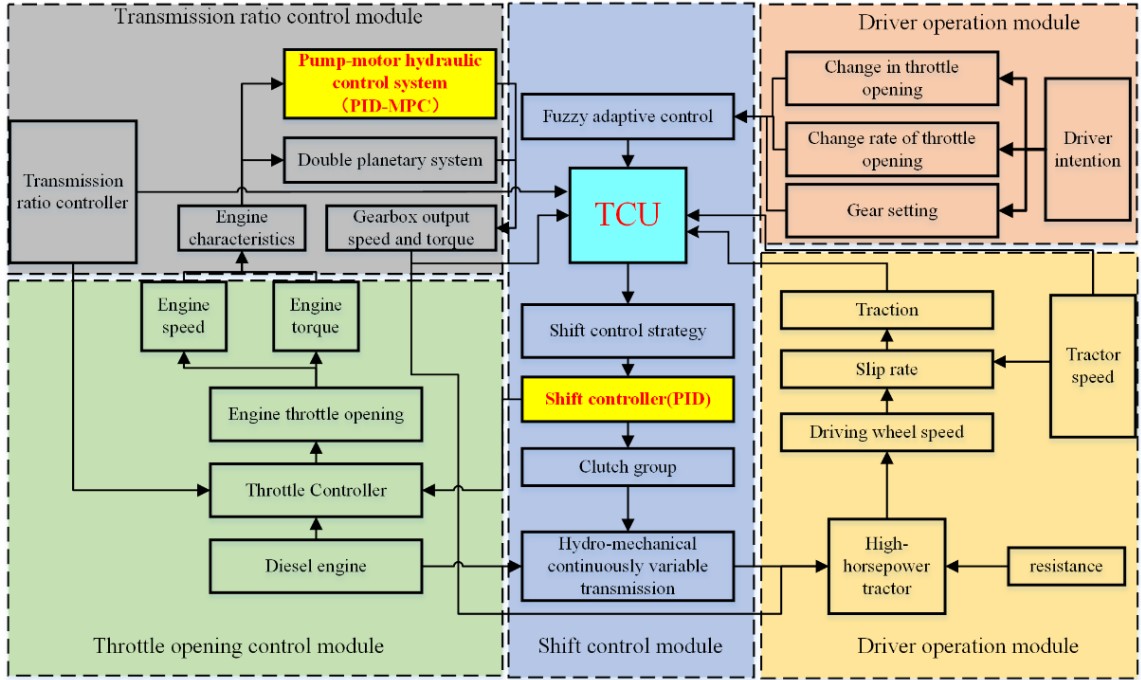

**Figure 8.** HMCVT shift control strategy.

### 3. Results and Discussion

*3.1. Test Verification of HMCVT*

3.1.1. Test Verification of Clutch Control System

Due to the nonlinear, time-varying and hysteresis characteristics of the wet clutch pressure control, this study uses PID control. The simulation of the control method is shown in Figure 9. It can be seen that the clutch system has a good control effect in step and sine wave signals. A single clutch test bench was established to verify the control effect as shown in Figure 10a. Because of the delay of the hydraulic system, the actual test is delayed for a short time under the control of step and sinusoidal signals (Figure 10b,c), but the overall control performance is in close agreement with the experimental measurements. The test results show that PID can satisfy the control requirements.

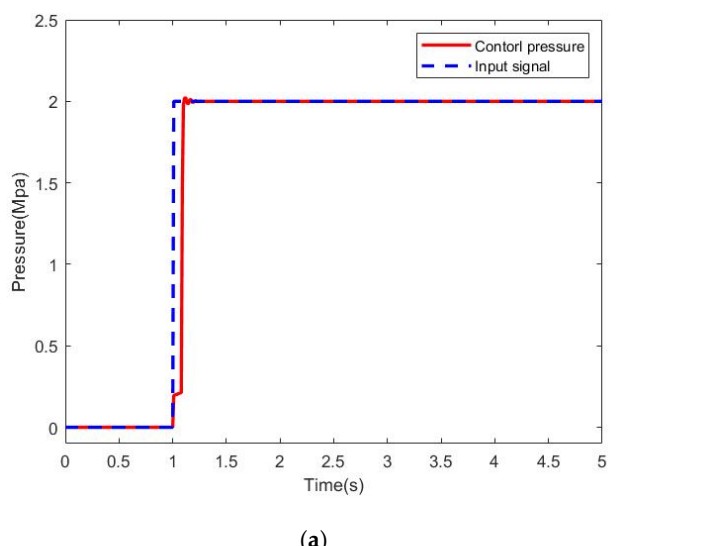
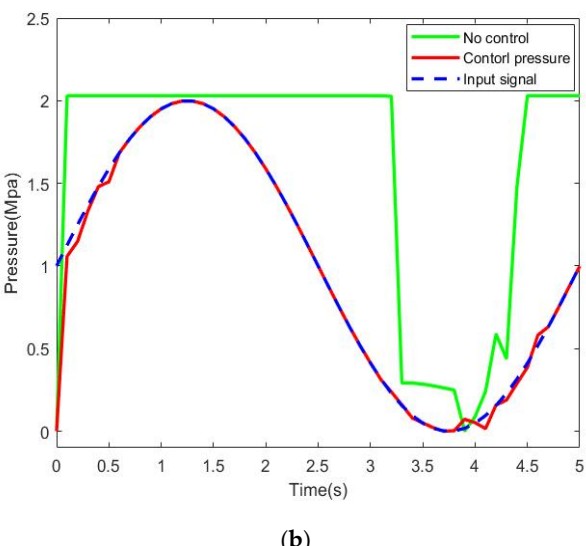

(**a**)                                                          (**b**)

**Figure 9.** Clutch response curve under different control signals. (**a**) Step signal control; (**b**) Sinusoidal signal control.

3.1.2. Test Verification of Variable Pump-Fixed Motor Control System

The variable pump-fixed motor hydraulic test platform was designed and fabricated (Figure 11a). The MPC method is used for experimental verification and simulation comparison, as shown in Figure 11b. It is very obvious that the simulated speed change is consistent with the experimental speed change, which proves the correctness of the designed variable pump-fixed motor hydraulic system model.

In this study, some control methods were used to analyze two operating conditions (constant displacement ratio and variable displacement ratio), which are shown in Figures 12 and 13. Figure 12 shows the simulation test result of the constant speed control under the state of constant displacement ratio. The variable pump-constant motor hydraulic system is controlled by the PID-MPC dual-loop control method. In the constant displacement ratio working condition, the 1s step signal is used for test, and the sinusoidal interference signal is added at 6s as a disturbance, which can observe the signal tracking performance and anti-interference performance of several control methods. Figure 12 shows that the PID-MPC dual-loop control method has a small overshoot compared to the PID control. Moreover, the control time for stable rotation speed is similar to PID. Compared with MPC, the PID-MPC dual-loop control method has a faster response speed. When interference signals are inserted, this control method shows excellent anti-interference ability. Therefore, the PID-MPC double-loop control inherits the advantages of fast PID response speed and a small MPC overshoot range in the speed control state. The constant displacement ratio torque change is shown in Figure 13. Under the PID-MPC double-loop

control, the torque of the pump-motor hydraulic system has the smallest fluctuation range and impact degree compared to PID and MPC independent control. The above shows that the control method has superior torque characteristics.

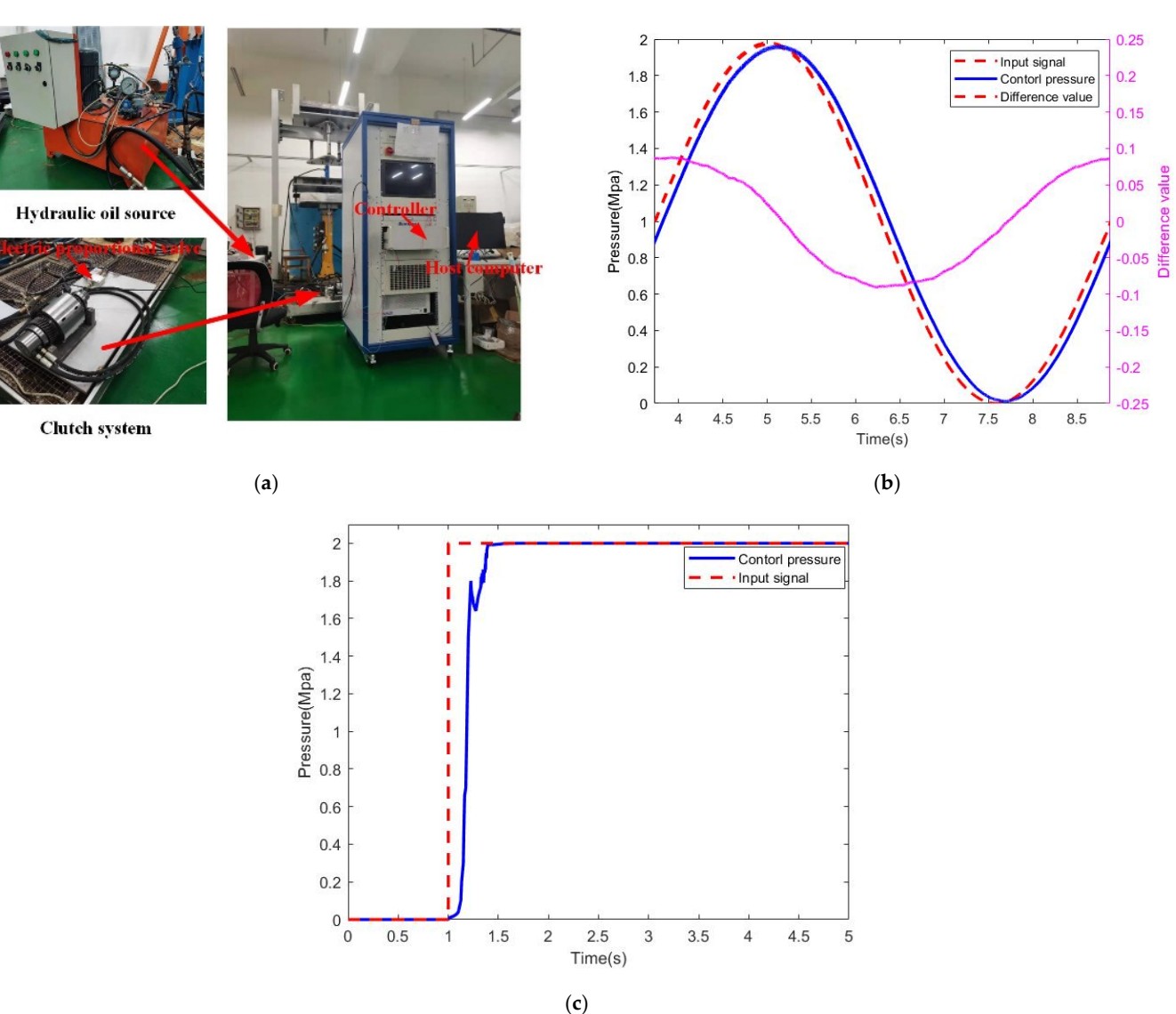

**Figure 10.** Results of clutch control test. (**a**) Clutch test bench; (**b**) Step signal control; (**c**) Sinusoidal signal control.

When the pump-motor hydraulic system is in variable displacement ratio, the performance of speed and torque are shown in Figure 14. Due to the predictive characteristics of MPC, the speed tracking performance has a certain delay in the first stage. Therefore, the speed tracking characteristics are poor in the starting stage of the pump-motor hydraulic system (Figure 14a). Although the speed tracking effect of PID control is good, its speed fluctuates wildly at 0 displacement ratio, and the speed stability is poor, which may exacerbate the instability of speed and the degree of loss in mechanical components. The PID-MPC double-loop control not only has a slight speed tracking error but also has the smallest speed fluctuation and impact at 0 displacement ratio. As shown in Figure 14b, the torque controlled by PID-MPC has the shortest time and the smallest amplitude change in the time domain, which results in the smallest torque impact. Therefore, PID-MPC dual-loop control has excellent tracking characteristics and stability. In summary, the PID-MPC control designed in this study is more suitable for the comprehensive control of the speed and torque of the pump-motor hydraulic system.

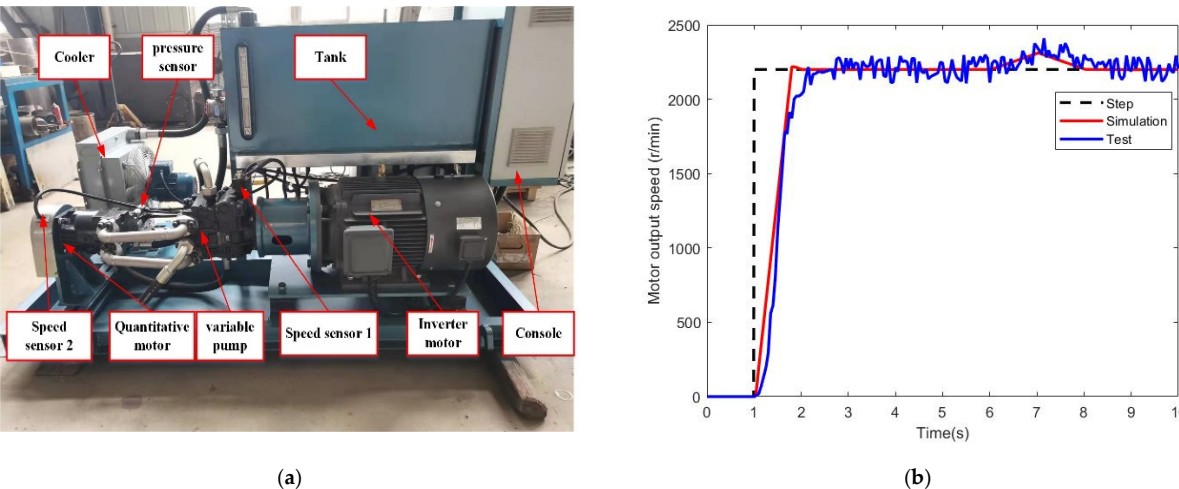

(**a**)                                                                    (**b**)

**Figure 11.** Pump-Motor test verification. (**a**) Variable pump-fixed motor hydraulic test platform; (**b**) Step experiment verification.

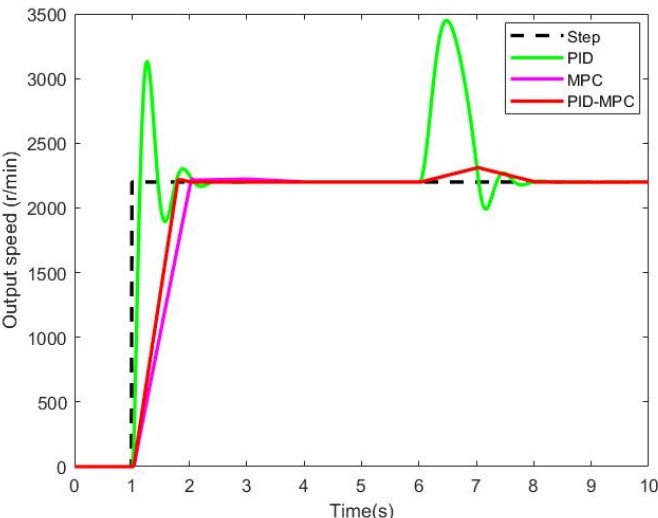

**Figure 12.** The change of motor control speed under step signal.

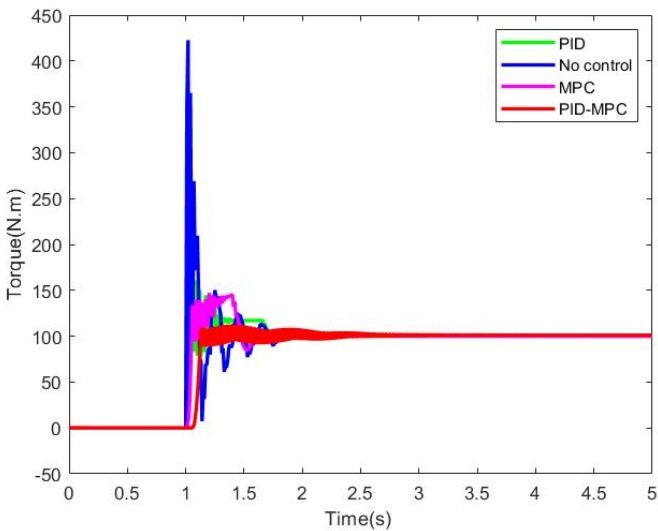

**Figure 13.** Variation diagram of motor control torque under step signal.

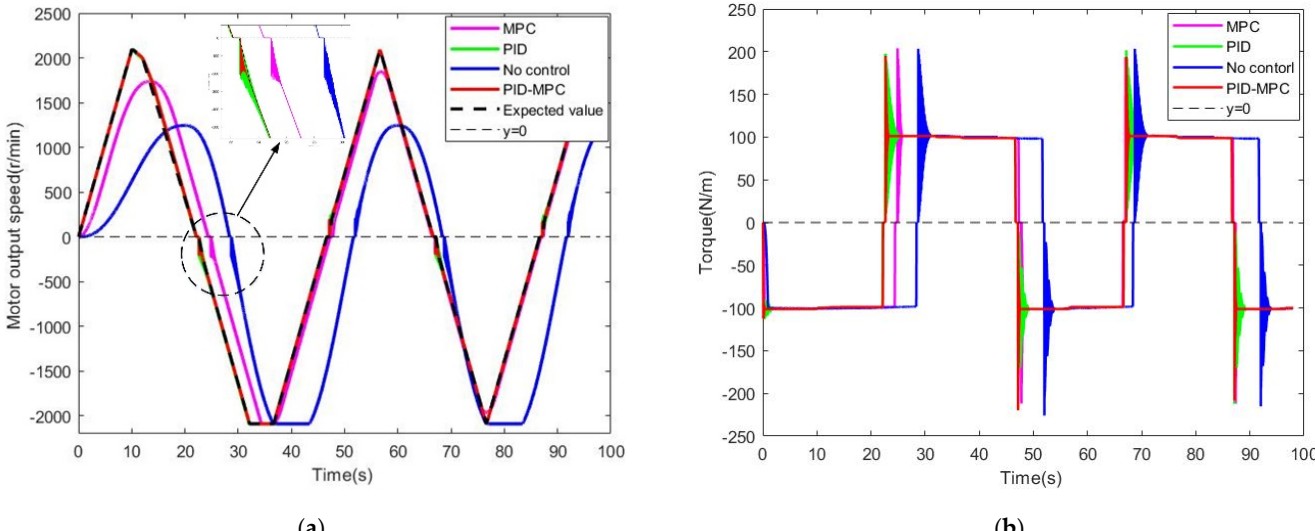

**Figure 14.** Speed and Torque control effect diagram at variable displacement ratio. (**a**) motor output speed; (**b**) motor output torque.

### 3.2. Shift Quality Evaluation Index

Generally, the control requirements for shifting are fast-shifting time, a small jerk and a stable transmission output speed [9]. However, because the essential attribute of The HMCVT is a continuously variable transmission and the control process time of shifting is short, the most important indicator for evaluating the quality of shifting is the fluctuation of output speed [21]. In this paper, the jerk (*J*) is used to represent the fluctuation of speed as an evaluation index of shifting quality. The smaller the value, the better the smoothness of shifting and the better the comfort of the tractor.

$$J = \frac{d\alpha}{dt} = \frac{d}{dt}\left(\frac{dv}{dt}\right) \tag{29}$$

The small jerk values mean minimal speed fluctuations in shifting gears and good ride comfort, which represents the degree of tractor comfort.

### 3.3. Definition of Control Strategy

The HMCVT is simulated and modeled according to the dynamic model in Section 2. The main parameters of the tractor are shown in Table 1. The HMCVT shift control strategy is different from other gearboxes (shifting only by the clutch) [22]. The HMCVT shift control strategy requires joint control of the pump-motor hydraulic system and the solenoid valve-clutch system. Its primary function is to control the displacement ratio of the pump-motor hydraulic system and the change sequence in the oil pressure of the clutch system. The control strategy is defined as follows:

$$H = (x, y, N) \tag{30}$$

where *x* represents the time difference between the maximum displacement ratio and the separation oil pressure of the clutch, and *y* represents the time difference between the clutch oil flushing and the oil draining. *N* = 0 means no control, *N* = 1 means in control state.

**Table 1.** The basic parameters of tractor.

| Tractor Model | Parameter |
|---|---|
| Maximum speed | 50 km/h |
| Engine rated power | 200 hp/149.1 kw |
| Engine rated speed | 2100 r/min |
| Maximum mass | 10,500 kg |
| Driving wheel tire | 480/80R50 |

### 3.4. Simulation of Integrated Control Shift Control Strategy

The jerk is obtained by adjusting the clutch engagement time to match the displacement ratio of the pump-motor hydraulic system under uncontrolled conditions (Figure 15), which is regarded as the most important indicator for evaluating the shifting stability of the tractor. It shows that the acceleration shock of the tractor is minimal at the maximum displacement of the pump-motor hydraulic system ($H = (0,0,0)$). Both early and late shifts at maximum displacement have great shock and poor stability. Especially when the gear shift is delayed, the reverse jerk of the tractor is large, so it is necessary to select the shift point at the maximum displacement of the pump-motor hydraulic system under no control conditions.

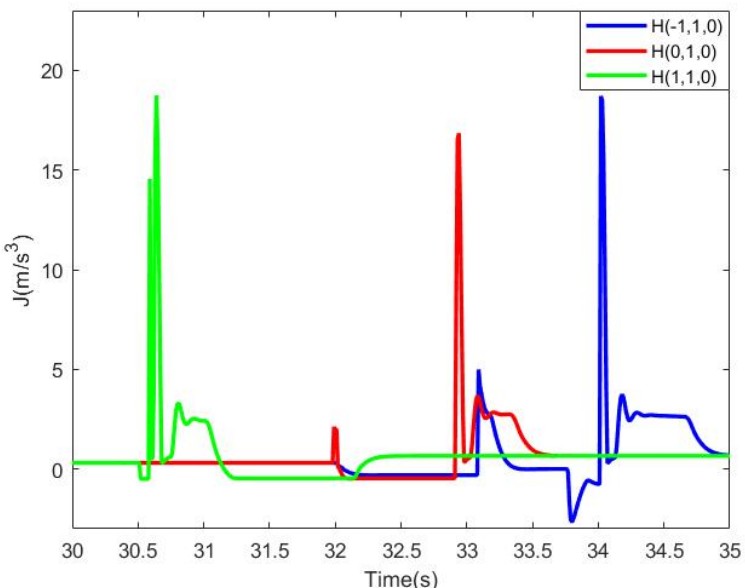

**Figure 15.** The jerk in uncontrolled strategies ($H = (x,1,0)$).

The results under the different control strategies are shown in Figure 16. It shows that the fluctuation of jerk is different in various control strategies. The jerk is the minimum value of about 7.9 m/s³ when $H = (0,1,1)$, and during the delay control ($H = (-1,1,1)$), the fluctuation amplitude of the shift shock is the largest. Moreover, it is very obvious that the maximum value of jerk with control is smaller than that without control (Figure 15). For example, when the HMCVT is controlled at the transition point ($H = (0,1,1)$), its jerk is about 53.5% smaller than the jerk without control ($H = (0,1,0)$), which shows better transition smoothness. Therefore, the HMCVT system exhibits better shift smoothness in the shift control strategy. It is well known that if the shift time is too short, it requires a larger shift torque per unit time, so the resulting shift jerk is higher. The frequent shifting may cause excessive overlap in clutch and pump-motor fluid system matching; therefore, it may cause shifting instability.

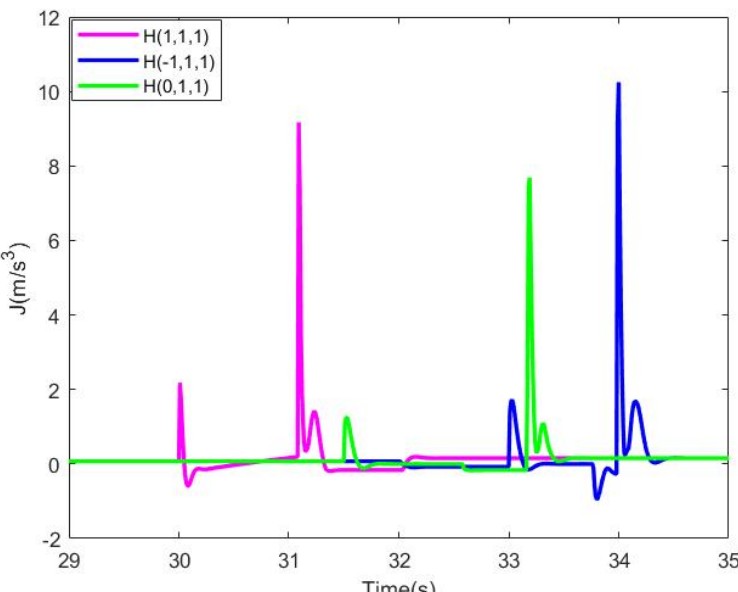

**Figure 16.** The jerk in different control strategies ($H = (x,1,1)$).

Therefore, a reasonable shifting time can reduce the jerk and improve the comfort of the tractor during shifting. In this study, a comprehensive control strategy (Equation (30)) is used to adjust the shift combination time, and the jerks with different control strategies are shown in Figure 17. There are $H = (0,1,1)$, $H = (0,1.5,1)$, $H = (0,2,1)$, $H = (0,2.5,1)$ four control optimization strategies, which show shifting at maximum displacement ratio. Figure 17 shows the variation of the extreme value of jerk for six different control strategies. The jerk degree of the optimized comprehensive control strategies is generally smaller than that of control strategies ($H = (1,1,1)$, $H = (−1,1,1)$). When the clutch engagement is controlled, the tractor speed gradually becomes smooth as the response time of the control clutch increases. However, when the control time exceeds a certain limit, the speed change rate suddenly increases ($H = (0,2.5,1)$), which is represented by an increase in the magnitude of the shift shock. Although the pump-motor hydraulic system is always in working condition, the excessively long coupling time causes a power interruption between the two clutches, which results in drastic changes in tractor speed and a sharp drop in shifting quality. Therefore, this phenomenon should be avoided as much as possible in tractor operation and driving. It can be seen from Figure 17 that the jerk fluctuation is the smallest when the comprehensive control strategy is $H = (0,2,1)$, which is about 6.05 m/s$^3$. Compared with the acceleration jerk when $H = (0,1,1)$ with control, it is reduced by 23.4%, and compared with the acceleration jerk when $H = (0,1,0)$ without control, it is reduced by 64.4%.

Other jerks are shown in Table 2 under different control strategies. In summary, when the HMCVT is in control strategy $H = (0,2,1)$, the speed change of the HMCVT tractor is the most stable, the acceleration shock of the tractor is the smallest, and the transition smoothness is the best. Figure 18 shows the changes in motor output speed and clutch torque when shifting gears. It can be seen that the clutch will have a specific impact on the output speed of the motor when the clutch is switched at the maximum displacement.

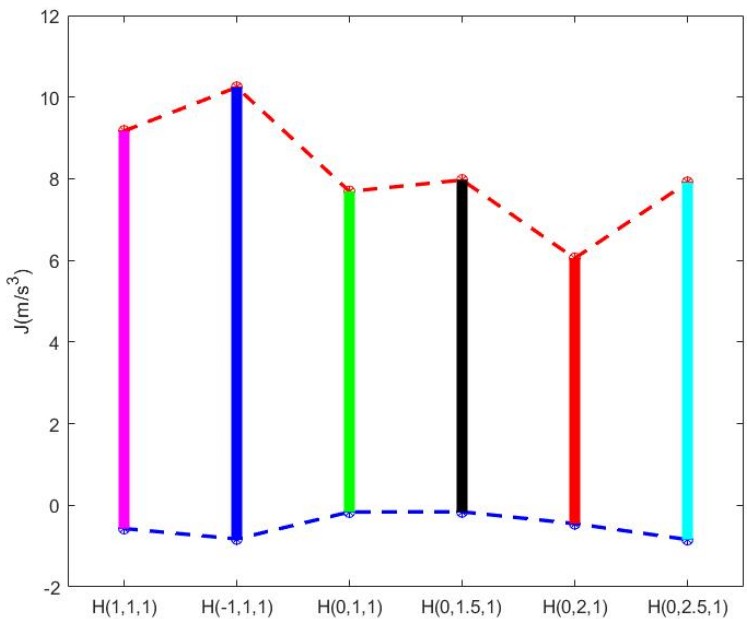

**Figure 17.** Amplitude change of the jerk in different control strategies (*H* = (x,y,1)).

**Table 2.** Jerk corresponding to different control strategies.

| Control Strategy | Max-Jerk (m/s³) | Control Strategy | Max-Jerk (m/s³) |
|---|---|---|---|
| *H* (0,1,1) | 7.81 | *H* (1,2,1) | 8.21 |
| *H* (0,1.5,1) | 8.14 | *H* (1,2.5,1) | 9.25 |
| *H* (0,2,1) | 6.27 | *H* (1,3,1) | 16.33 |
| *H* (0,2.5,1) | 8.12 | *H* (−1,1,1) | 10.55 |
| *H* (0,3,1) | 14.44 | *H* (−1,1.5,1) | 10.27 |
| *H* (1,1,1) | 9.36 | *H* (−1,2,1) | 9.64 |
| *H* (1,1.5,1) | 8.61 | *H* (−1,2.5,1) | 12.79 |

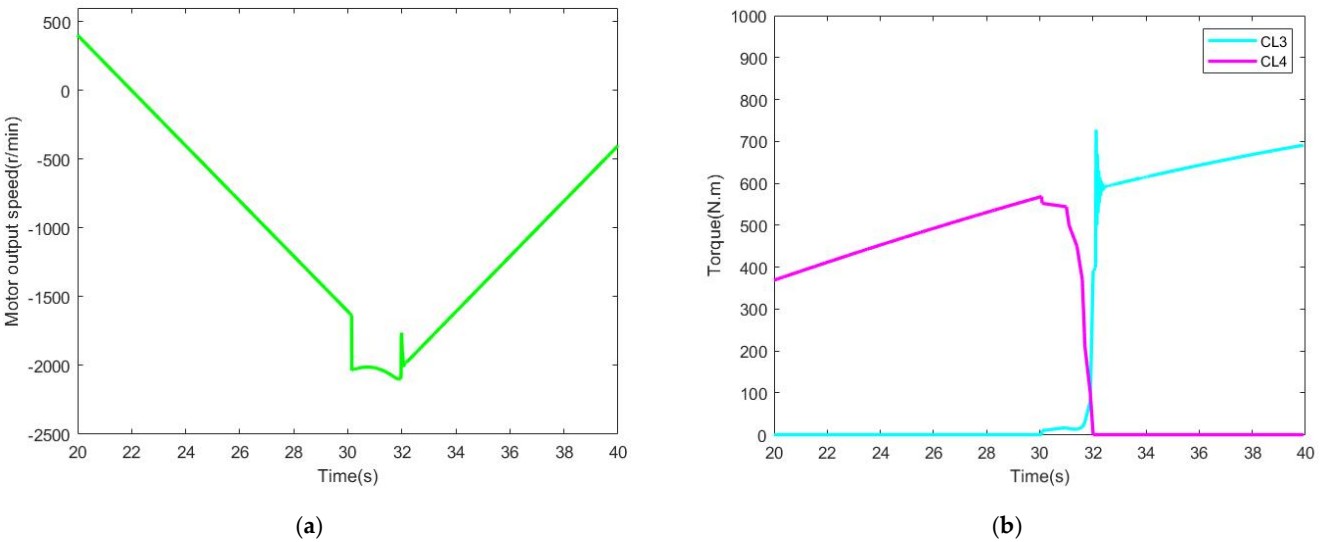

**Figure 18.** Changes in motor speed and clutch torque when shifting. (**a**) Motor output speed; (**b**) Clutch torque change.

## 4. Conclusions

The critical components of the HMCVT, including variable pump-fixed motor hydraulic system, clutch system and mechanical shafting, are modeled in detail in this article.

The PID-MPC dual-loop control is proposed for the variable pump-fixed motor hydraulic system of the HMCVT. The inner loop controls the displacement ratio, and the outer loop controls the motor speed through feedback, which overcome the inherent delay of the hydraulic system. The double-loop control improves the speed tracking performance and anti-interference ability of the pump-motor system, increases the robustness of the system, and realizes the precise control of the output speed of the precise hydraulic system. The clutch system is verified by PID control, and the tracking control simulation and test results show that PID can have excellent tracked performance under both phase and sinusoidal signals.

A comprehensive control strategy for HMCVT shifting is proposed, which controls the shifting process of the tractor for the overall level and adjusts different control parameters to obtain the optimal shifting performance. Compared with no control, the comprehensive control optimization strategy shows better shift smoothness, reduces the shift jerk by about 64.4%, and improves the shift quality.

**Author Contributions:** Conceptualization, J.L., B.H., Y.Z. and Z.Z.; methodology, J.L., B.H. and Z.Z.; software, J.L.; validation, J.L., H.D. and B.H.; formal analysis, J.L., B.H., Y.Z. and Z.Z.; investigation, B.H.; resources, Z.Z.; data curation, J.L., B.H., H.D. and Z.Z.; writing—original draft preparation, J.L.; writing—review and editing, J.L. and B.H.; funding acquisition, Z.Z. All authors have read and agreed to the published version of the manuscript.

**Funding:** The authors would like to thank the National Natural Science Foundation of China under Grant (No. 52072407).

**Institutional Review Board Statement:** Not applicable.

**Informed Consent Statement:** Not applicable.

**Data Availability Statement:** Not applicable.

**Conflicts of Interest:** The authors declare that they have no known competing financial interests or personal relationships that could have appeared to influence the work reported in this paper.

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
