# Peer review of "Designing Comprehensive Shifting Control Strategy of Hydro-Mechanical Continuously Variable Transmission"

_applsci, doi:10.3390/app12115716_

Round 1
Reviewer 1 Report
The paper is well written in all chapters. I notice the mathematical models presented and the development of an experimental part for a continuous transmission.
I recommend checking the work carefully, I notice small leaks, for example relationship 21 is numbered 2 times.
Author Response
Responses to Comments and Suggestions
Dear reviewer,
Thanks for the reviewer’s approval and concern for our manuscript. We are sorry for making the mistakes in equation number. Those comments are all valuable and very helpful for revising and improving our paper, as well as the important guiding significance to our research. We have studied comments carefully and have made correction. We have carefully checked the manuscript and proofread every equation, figure and table. We have modified the relationship in response to the problems you raised,(Please see the attachment.)
We tried our best to improve the manuscript and made some changes in the manuscript.
Once again, we appreciate for reviewers’ warm work earnestly, and hope that the revised version will meet with approval.
Thank you and best regards.

Reviewer 2 Report
The manuscript’s (MS) topic deals with the investigation on shifting control strategy of HMCVT in a high horsepower tractor for enhancing shift quality. Mathematical model of the transmission assembly components such of the HMCVT is established to analyze the exact shifting process.
A thing to take into consideration is that similar systems, has already been presented in previous international publications. But, using the PID-MPC double-loop control can be interesting for readers of journal of Applied Sciences. I believe scientific evidence of MS is enough to be published in a high standard scientific journal as Applied Sciences. Therefore, it is acceptable for publication after some major revisions as follows:
The manuscript should be organized including the sections of “Abstract”, “Keywords”, “Introduction”, “Materials and Methods”, “Results”, “Discussion” and “Conclusions”. The sections of “Materials and Methods”, “Results”and “Discussion” are missing. In some chapters, “Material and Methods” and “Results and Discussion” are intertwined and mixed. These sections should be separated from each other and presented separately. The current structure of the MS is not suitable for a scientific article.
Title: A shifting control strategy of HMCVT was investigated in a high horsepower tractor in MS. Therefore, the term of “tractor” should be impressed in title.
Abstract: Avoid using the abbreviations in abstract
Keywords: Add “tractor” and avoid using the abbreviations
I think, the lines between 98-104 includes the revision notes of a reviewer or part of the article template file. I think it was added here by mistake. It should be removed.
The text in the figures is too small, making it difficult to read.
Figure 17 is the repetition the data in Figure 15 and 16. One data should be presented one time.
All tables and figures should be mentioned in the text. Please check the referring for all tables and figures. I can’t find any sentence refered to Figure 19.
Author Response
Responses to Comments and Suggestions
Dear reviewer,
Thank you for your comments concerning our manuscript. The comments are all valuable and very helpful for revising and improving our paper, as well as the important guiding significance to our research. We have studied comments carefully and have made correction which we hope to meet with your approval.
We have made changes for the manuscript, as shown in the updated version. In the updated version, the changes are marked up using the “Track Changes” function (according to the editor's request). The following are reply to the reviewer:
Main comments:
Comment 1: The manuscript should be organized including the sections of “Abstract”, “Keywords”, “Introduction”, “Materials and Methods”, “Results”, “Discussion” and “Conclusions”. The sections of “Materials and Methods”, “Results” and “Discussion” are missing. In some chapters, “Material and Methods” and “Results and Discussion” are intertwined and mixed. These sections should be separated from each other and presented separately. The current structure of the MS is not suitable for a scientific article.
Response 1: Thanks for the expert’s advice. The structure part of the manuscript is revised based on comment of reviewer. The modeling of the various modules of HMCVT and the design of the control system are filled in the second section (mathematical modeling and methods). The experimental verification of each component module of HMCVT (variable pump-constant motor hydraulic system and clutch system) is filled in the first part of the third section (results and discussion), which is used as test certificate of the second part (building model and designing control system). The latter part of the third section is to simulate the control strategy and analyze the results. The structure directory of the revised manuscript is shown in the attachment. Please see the revised manuscript for details.
Comment 2: Title: A shifting control strategy of HMCVT was investigated in a high horsepower tractor in MS. Therefore, the term of “tractor” should be impressed in title.
Response 2: Thanks for the expert’s advice. The title of the manuscript has been revised “Designing Comprehensive Shifting Control Strategy of Hydro-mechanical Continuously Variable Transmission for the high horsepower Tractor”.
Comment 3: Avoid using the abbreviations in abstract Keywords: Add “tractor” and avoid using the abbreviations
Response 3: Thanks for the expert’s advice. Abbreviations in abstract and keywords have been revised. The “tractor” has been added to the keyword.
Comment 4: I think, the lines between 98-104 includes the revision notes of a reviewer or part of the article template file. I think it was added here by mistake. It should be removed.
Response 4: Thanks for the expert's advice. We are sorry for making the mistakes. The lines (between 98-104) which includes the revision notes of a reviewer or part of the article template file has been removed.
Comment 5: The text in the figures is too small, making it difficult to read.
Response 5: Thanks for the expert's advice. We are sorry for making text in figures too small. Figures in the manuscript are resized to make the text clearer, which is more convenient for reviewers and readers to understand the meaning of Figures
Comment 6: Figure 17 is the repetition the data in Figure 15 and 16. One data should be presented one time.
Response 6: Thanks for the expert's advice. To avoid data duplication, Figure 17 is removed.
Comment 7: All tables and figures should be mentioned in the text. Please check the referring for all tables and figures. I can’t find any sentence refered to Figure 19.
Response 7: Thanks for the expert's advice. We have checked the referring for all tables and figures. All tables and figures are described in the manuscript. For example, Figure 19 is described in the last paragraph of Part 4 “Fig.19 (Fig.18 after correction) shows the changes in motor output speed and clutch torque when shifting gears. It can be seen that the clutch will have a specific impact on the output speed of the motor when the clutch is switched at the maximum displacement”.
The above are our team's answers to the experts' questions. Once again, we express our respect for the expert. We very much hope that these explanations will enable the expert and readers to further understand our research work.
We tried our best to improve the manuscript and made some changes in the manuscript.
Once again, we appreciate for reviewers’ warm work earnestly, and hope that the revised version will meet with approval.
Thank you and best regards.

Reviewer 3 Report
The paper is interesting, is substantially theoretically and applied, and meets the scientific requirements to be published.
Author Response
Responses to Comments and Suggestions
Dear reviewer,
Thanks for the reviewer’s approval and concern for our work. And we have revised and improved the manuscript further, which enable the expert and readers to further understand our research work.
Once again, we appreciate for reviewers’ warm work earnestly and approval.
Thank you and best regards.

Round 2
Reviewer 2 Report
Dear Authors
I thank you for precise work while revising the manuscript and detailed responses to reviewers. All my comments and suggestions have been fulfilled. Therefore I think your article is ready for publication.